# The Potential Role of Exosomes in Communication Between Astrocytes and Endothelial Cells

**DOI:** 10.3390/ijms26104676

**Published:** 2025-05-14

**Authors:** Joanna Czpakowska, Andrzej Głąbiński, Piotr Szpakowski

**Affiliations:** Department of Neurology and Stroke, Medical University of Lodz, Zeromskiego 113 Street, 90-549 Lodz, Poland; joanna.czpakowska@umed.lodz.pl

**Keywords:** exosomes, extracellular vesicles, astrocytes, endothelial cells, blood–brain barrier, neurodegeneration, neuroinflammation, miRNA, intercellular signaling, CNS

## Abstract

Exosomes are extracellular vesicles secreted by almost all types of cells. Their release allows for the transport of specific regulatory cargo into the recipient cells and the modulation of their activity. Vesicular communication has also been identified as an important mechanism for the regulation of numerous cellular activities in the brain tissue, contributing to proper neuronal functions and brain homeostasis. In this work, we focus on the role of exosomes and extracellular vesicles in the communication between astrocytes and brain endothelial cells, two major components of the blood–brain barrier. We perform a comprehensive review of the latest studies highlighting the role of exosomes in astrocyte-endothelial cell crosstalk within the blood–brain barrier. We have also described the role of particular exosomal miRNAs in the regulation of astrocytes and brain endothelial cell functions, and discuss some future implications.

## 1. Introduction

### 1.1. The Blood–Brain Barrier Role in CNS

The blood–brain barrier (BBB) is a complex structure that fulfills an essential role in maintaining homeostasis and providing protection to the central nervous system (CNS) [1,2]. The BBB present in cerebral capillaries enables the exchange between blood and the brain simultaneously restricting the entry of unfavorable factors [1,3]. Microvascular endothelial cells (ECs) constitute the main cellular component of the BBB. Astrocytes—particularly their end-feet—and pericytes also contribute to the BBB function [1,4]. The cells comprising the BBB, together with the capillary basement membrane, microglia, and neurons, form the neurovascular unit (NVU) [1,5]. The BBB remains tight if CNS homeostasis is preserved. Conversely, the BBB is involved in maintaining these conditions in its environment by acting in diverse ways. The major one is based on utilizing ion channels and transporters such as Na^+^, K^+^-ATPase. These channels allow for maintaining specific concentrations of ions, different from the levels found in plasma, which is essential for neuronal and synaptic activity. The transporters specialized in the regulation of neurotransmitter levels restrict amino acid flow in CNS, therefore preventing the harmful consequences of their altered levels [1,6]. Additionally, the impermeability of the BBB prevents the entry of other potentially detrimental substances such as macromolecules or neurotoxins, which could cause irreversible damage to the CNS [1,7].

### 1.2. Importance of Endothelial Cells in the BBB

ECs serving as the crucial element of the BBB distinguish themselves from regular ECs. Their characteristics allow for fulfilling specific functions in the brain [8,9]. Primarily, ECs present in the CNS have developed distinctive tight junctions (TJs) responsible for preventing paracellular transport [9,10]. This feature provides tightness of the BBB. TJs, present in ECs aside from CNS, are characterized by three types. The most common are present at the apical area, and on the basolateral part are situated desmosomes. In addition to desmosomes, adherens junctions are also present. Desmosomes are absent in ECs from CNS; additionally, their junctions are distributed dissimilarly [9]. TJs consist of membrane proteins which are occludin, junctional adhesion molecules (JAMs), and claudins [11,12]. These transmembrane proteins play an essential role in maintaining TJs while simultaneously interacting with proteins in the cytoplasm. The membrane-associated guanylate kinase (MAGUK) family proteins are the group of most important cytoplasmic proteins associated with the transmembrane proteins. The MAGUK group consists of zonula occludens-1 (ZO-1), ZO-2, and ZO-3 [12,13]. Their characteristic feature is the presence of the PDZ domain, which allows for interaction with occludin, JAMs, and claudins. The proteins connected by MAGUK members are, in turn, anchored to the actin cytoskeleton [12].

Specific transporters are another distinguishing property of ECs present in the BBB [14]. Most of them require an increased amount of ATP since they are ATP-binding pumps. Due to this fact, mitochondria in ECs from CNS are even five times more abundant compared to the peripheral ECs [9]. This type of transporters belongs to the ATP-binding cassette transporter (ABC) superfamily, which, with the solute carrier (SLC) superfamily, constitutes two of the most important groups [15,16]. ABC transporters are responsible for a selective efflux of substances towards blood which is possible due to the energy obtained after ATP hydrolysis. Conversely, SLC allows for the passive transport of smaller particles [16,17]. To the best-known ABC transporters belong P-Glycoprotein (ABCB1/P-gp) which prevents the influx of undesired drugs into the brain, Breast Cancer Resistant Protein (ABCG2/BCRP) which causes the efflux of toxic molecules, Multidrug Resistance Protein 1/2 (ABCC1/MRP1/2) which acts similarly, and ATP binding cassette subfamily G member 4 (ABCG4) which responsible for cholesterol transport [16,18,19,20,21]. Among the most studied SLC there can be distinguished Glucose Transporter 1 (SLC2A1/GLUT1), Sodium-dependent lysophosphatidylcholine symporter 1 (SLC59A1/Mfsd2a), Organic Anion Transporter Polypeptide 1A2 (SLCO1A2/OATP1A2), L-Type Amino Acid Transporter 1 (SLC7A5/LAT1), and Solute Carrier Family 22 Member A3 (SLC22A3/OCT3) [16,22,23,24,25].

### 1.3. Importance of Astrocytes in the BBB

Astrocytes represent another critical component of the BBB. Their location in NVU in the proximity to both neurons and ECs allows them to affect these cells and thus influence the condition of the BBB [26,27]. Their specialized end-feet cover the cells of NVU and modulate them thanks to the increase in Ca^2+^ levels in these structures [28]. Astrocytes serve as a bridge between neurons and blood vessels, allowing bidirectional signal exchange. This is evident when blood flow is controlled by glutamate-mediated signals resulting from the neuronal secretion of nitric oxide and arachidonic acid derivatives from astrocytes [26,29]. Astrocytes are also capable of inducing the contraction and dilation of vascular smooth muscle cells, thereby affecting cerebral blood flow and neuronal function [30]. This mechanism is possible due to the presence of inter-astroglial gap junctions which contribute to the more efficient communication between cells, making astrocytes a coordinated aggregate [28,31]. The mentioned properties allow astrocytes to balance pH in the NVU environment, distributing to neurons the building blocks of neurotransmitters, and causing their breakdown [26,32].

Astrocytes also contribute to ensuring the BBB proper characteristics such as maintained TJs, the presence of specific enzymes, and the efficient functionality of transporters in concordance with their intended use [26]. Studies on blood vessels derived from outside CNS showed that astrocytes can influence them to develop characteristics of the BBB. Additionally, the analysis of astrocytes’ secretomes confirms their role in sustaining the proper condition of the BBB and its regulation [30,33]. In a study on mice genetically deprived of astrocytes, it was also confirmed that astrocytes are indispensable in maintaining the BBB. The results of the study indicate that astrocyte ablation leads to the worsening of the already present BBB dysfunctions, and contributes to the malfunction of TJs. Additionally, the loss of astrocytes will not trigger the proliferation of the nearby cells which will negatively impact long-term BBB stability [34]. The study on mice conducted by Jackson RJ et al. reveals that astrocytes influence the tightness of the BBB by the secretion of Apolipoprotein E4 (ApoE4). *APOE2*, *APOE3*, and *APOE4* constitute the three isoforms of the *APOE* gene. ApoE4 is identified as an unfavorable factor in maintaining the integrity of the BBB. Its activity leads to the loss of the BBB tightness due to improper functioning of TJs. Additionally, the level of matrix metalloproteinase-9 (MMP9) increases in the presence of ApoE4, and therefore, the breakdown of the BBB intensifies. ApoE4 also contributes to the reduced astrocytic end-foot coverage of cerebral blood vessels [35].

## 2. Interactions Between ECs and Astrocytes

Both ECs and astrocytes are directly involved in the regulation of the BBB functioning due to their specific physiology and abilities (Figure 1). Because of their location in proximity to each other, these cells developed mechanisms of communication and mutual interactions.

Astrocytes can influence ECs by the paracrine secretion of factors that act through various signaling pathways. One of them is the Hedgehog (Hh) pathway. Astrocytes produce Sonic Hh (SHh) protein which is the agonist of the Patched-1 receptor present on the surface of ECs. SHh activates Patched-1, initiating intracellular signal transduction. As a result, TJ proteins are expressed, enabling ECs to acquire BBB characteristics. Astrocytes can also release retinoic acid, which, by the activation of retinoic acid receptor β (RAR-β) on Ecs, contribute to the production of ZO-1 and vascular endothelial cadherin (VE-cadherin) proteins making up TJs and transporters. Additionally, retinoic acid has the ability to influence the Hh pathway [30,36]. Astrocytic laminins constitute another vital aspect of maintaining TJ proteins [37]. Additionally, the activation of the Hh pathway causes a shift in transendothelial electrical resistance and improves the BBB tightness by intensified expression of claudin-3, -5, occludin, (Junctional Adhesion Molecule A) JAM-A, VE-cadherin, and laminin. Astrocytes can also induce properties of vascular endothelium by secretion of vascular endothelial growth factor (VEGF) and angiopoietins (Ang1) which additionally contributes to the increase in the BBB tightness. SHh activity also leads to constricting the adhesion and transmigration of cells from the immune system by limiting the expression of intercellular adhesion molecule (ICAM)-1 in ECs [30,36,38,39]. Additionally, Ang1 activates mitogen-activated protein (MAP) kinase which provides safety for the BBB and preserves it from unfavorable agents. Activation of phosphoinositide 3-kinase (PI3-kinase) by Ang1 leads to reduced EC apoptosis [30,36,39].

The impact of ECs on astrocytes was also investigated in a few studies. The available data confirm their importance in astrocyte functioning. The studies conducted on astrocytes and EC co-cultures indicate the role of ECs in acquiring maturity by astrocytes and mediating their differentiation [40]. It was noticed that ECs influence the differentiation of astrocytes by the production of leukemia inhibitory factor (LIF) [41]. The impact of ECs on astrocytes in co-culture is also visible in the intensified Ca^2+^ responses in astrocytes after exposition to pain and inflammation triggers which imply the intensified intracellular signals transduction [40]. The results from another study on co-culture show that the amount of aquaporin-4 on the astrocytes’ end-feet increases in the presence of ECs [41]. The study conducted by Taylor X et al. revealed the importance of LPS-activated ECs in the induction of astrocyte neurotoxicity with preserved phagocytic properties. The activation of astrocytes takes place due to the elevated expression of complement component 3 (C3). Additionally, the expression of the *Dcn* gene encoding extracellular matrix protein Decorin increases which distinguishes this type of activation from the one mediated by microglia [42]. Hultman K et al. demonstrated that the co-culture of these two types of cells results in the increased expression of plasminogen activator inhibitor type-1 (PAI-1) in astrocytes. The elevated amount of PAI-1 results in the inhibition of tissue-type plasminogen activator (t-PA), and plasmin cannot break down fibrin [40].

## 3. Exosomes as the Strategy of Communication Between Astrocytes and ECs

The alternative form of communication and interaction between astrocytes and ECs may constitute extracellular vesicles (EVs) that can be released and targeted by every type of cell [37]. EVs can carry proteins; lipids; metabolites; RNAs, such as microRNA (miRNAs) and messenger RNAs (mRNAs); DNA; and organelle. The cargo of EVs can influence the functioning of cells, similarly as it takes place in paracrine communication [43].

Considering the distinct biogenesis and other characteristics of EVs, they are commonly divided into exosomes and microvesicles (Figure 2). The pathway of obtaining exosomes comprises endocytosis, the formation of early endosomes, late endosomes, and multivesicular bodies (MVBs) with intraluminal vesicles (ILVs) which release exosomes by exocytosis. MVBs are able to reach and fuse with the cell membrane due to the cytoskeletal transport comprising microtubules. There is also the possibility that MVBs will undergo degradation by lysosomes or autophagosomes [44,45,46]. The endosomal sorting complex required for transport (ESCRT) proteins contributes to the release of ILVs. ESCRT comprises ESCRT-0, -I, -II, and -III and vacuolar protein sorting 4 (Vps4) complexes which allow for the sorting of the MVBs content and their release. Other proteins involved in exosome secretion are Soluble NSF attachment proteins (SNARE) and Rab GTPases. Both types allow for MVBs docking and further fusion with cell membranes [44,45,47]. This type of EVs range in diameter from 30 to 100 nm making them the smallest kind. Exosomes are characterized by regularity in shape and size, which is the consequence of their organized mechanism of formation [48,49,50]. On their surface can be found markers such as CD81, CD9, and CD63 which are the most abundant proteins in exosomes [51]. Microvesicles’ biogenesis relies on a less complicated process which is the budding of the plasma membrane leading to their irregular shape and size. The diameter of obtained microvesicles extends from 50 nm to 1000 nm, but in some cases, they can reach 10 μm. Their specific surface markers are integrin-β, CD40, and selectins. The exterior of microvesicles may also contain proteins of the cells of their origin [48,49,50].

Exosomes draw particular attention in the context of neurological, especially neuroimmunological, diseases since their properties embrace alleviating inflammation, lowering the amount of reactive oxygen species (ROS), and maintaining homeostasis [52,53]. Their smaller size makes it easier to enter the recipient cells which leads to the more effective delivery of exosomal cargo and therefore, more efficient cellular responses [54]. The number of research on exosomes is increasing year by year and overtaking the number of publications about microvesicles which can be explained by the exosomal properties regarding use in potential therapies, the route of biogenesis, successful delivery to cells, and increased oral bioavailability [55]. Exosomes also became the subject of many clinical trials. The most common clinical applications of exosomes encompass biomarkers, exosome therapy, exosome drug delivery systems, and exosome vaccines. Usually, these exosomes are obtained from plant cells, and in the case of human origin, their sources are mesenchymal cells, dendritic cells, and T cells [56,57].

The studies on the mutual influence of astrocytes and ECs with the use of exosomes are restricted; therefore, there is a need for further research on this subject. Currently, there are available studies concerning the exosomes released by astrocytes and ECs that describe their characteristics, mechanisms of action, potential targets, and cargo profiles. The synthesis of the current knowledge regarding these exosomes in the CNS environment will provide a foundation for further research and enable the design of experiments concerning communication between astrocytes and ECs through exosomes.

### 3.1. Characteristics of Exosomes Secreted by Astrocytes

#### 3.1.1. Astrocyte-Derived Exosomes Impact on CNS Cells

The communication between astrocytes and the main part of CNS, which are neurons, relies mostly on astrocytic processes. Besides the typical way of sending and receiving signaling molecules, astrocyte end-feet secrete exosomes that fulfill the role of signal carriers (Figure 3) [58]. In the study conducted by Venturini A et al., astrocytic processes were obtained from the adult rat cerebral cortex and were maturing in a co-culture of neurons. After the analysis of the astrocyte-derived exosomes (AS-Exos), it was concluded that neurons constituted the only target of AS-Exos which implies their lack of randomness in cell targeting. The cells in the co-culture were fluorescently labeled. The confocal images showed cells containing glial acidic fibrillary protein (GFAP) which is the characteristic protein for astrocytes distinguishing these cells from others. Neurons were labeled with specific markers which are microtubule-associated protein 2 (MAP-2) and β III tubulin. AS-Exos were visible due to PHK67 dye. The uptake of AS-Exos by neurons was visible in the confocal images of the co-culture where their presence was noticed near the cells negative for GFAP and positive for MAP-2 and β III tubulin. AS-Exos were confirmed to be internalized by neurons as they were located near the nucleus. AS-Exos were also confirmed to carry neuroglobin (NGB) which is a neuron-specific protein having neuroprotective activities. Releasing NGB by astrocytes in exosomes indicates their involvement in maintaining neuronal homeostasis [59]. AS-Exos are also engaged in the formation of synapses by activating transforming growth factor β (TGF-β) signaling with the use of fibulin-2. This protein is present in AS-Exos in significantly higher amounts in comparison to other exosomes [60].

Oligodendrocytes constitute another important cell type of the CNS that can be influenced by AS-Exos. Especially oligodendrocyte progenitor cells (OPCs) are the subject of AS-Exos activity which comprises enhancing their differentiation into oligodendrocytes and supporting their migration. However, in the condition of hypoxia, AS-Exos act by suppressing OPC proliferation. The presented properties of AS-Exos indicate their potential application in myelin regeneration therapy [61].

AS-Exos also exhibit the ability to regulate microglia functioning. In the study concerning AS-Exos treated with morphine, it was concluded that after their uptake by microglia, Toll-like receptor 7 was activated, which resulted in their diminished phagocytosis. In the study with a mouse model of experimental autoimmune encephalomyelitis (EAE), it was pointed out that α-B-crystalline protein present in AS-Exos suppressed microglial activation and subsequent inflammation [62].

#### 3.1.2. AS-Exos Role in Neuronal Protection and Regeneration

The involvement of AS-Exos in neuroprotection and neuroregeneration is visible in studies concerning traumatic brain injury (TBI). The research performed by Chen W et al. indicates that astrocytes participate in neuronal repair by the release of exosomes with enclosed gap junction alpha 1–20 k protein (GJA1–20 k). This protein is involved in apoptosis suppression and enhancing mitochondrial activity, which contributes to neuronal protection and regeneration [63]. Similar observations were made in a study on ischemic stroke. This condition is able to trigger neuronal autophagy which is an unfavorable phenomenon leading to the excessive loss of these cells. In this study, researchers prove that neuronal autophagy is inhibited by AS-Exos [64]. Neuronal damage resulting from hypoxia, ischemia, and hypoglycemia is also prevented by prion protein (PrP). The study conducted by Guitart K et al. reveals that astrocytes cultured in the mentioned conditions exhibited higher amounts of PrP in their exosomes influencing the positive survival of neurons. Interestingly, the PrP released by neurons did not have such properties which emphasizes the importance of AS-Exos [65].

#### 3.1.3. AS-Exos Importance in Neurodegenerative and Neuroinflammatory Diseases

AS-Exos are thought to be involved in the course of neuroinflammatory diseases [66]. It was proven that AS-Exos contain an intermediate filament protein which is GFAP. The secretion of GFAP-positive AS-Exos was noticed to be intensified in response to the inflammation which confirms their role in pathological processes in CNS [67]. A similar observation was also made in our recent study regarding the AS-Exos effect on CD4^+^ T-cells obtained from multiple sclerosis (MS) patients. The results show that AS-Exos influence the secretory activity of T-cells and therefore, may affect the course of MS, which is a neurodegenerative and neuroinflammatory disease [68].

The importance of AS-Exos was observed in a few studies concerning Alzheimer’s Disease (AD), which is the most prevalent neurodegenerative disease. Winston CN et al. indicated that AS-Exos are useful in serving the role of predictive biomarkers for the development of AD. AS-Exos obtained from patients with mild cognitive impairment transforming into AD dementia were characterized to carry a higher amount of complement proteins in comparison to the subjects without the disease progression [69]. Chiarini A et al. draws attention to the mechanism of AD exacerbation in which AS-Exos take part. Amyloid-β (Aβ) and hyperphosphorylated Tau (p-Tau) oligomers constitute the two main pathological agents of AD. The binding of Aβ to calcium-sensing receptor (CaSR) on the surface of astrocytes leads to the intensified accumulation of p-Tau inside these cells. Next, p-Tau is secreted in AS-Exos contributing to the disease progress [70]. The cargo of AS-Exos derived from AD patients is also characterized by increased levels of β-site amyloid precursor protein-cleaving enzyme 1 (BACE-1) and γ-secretase. Conversely, the substance contributing to neuronal survival which is the glial-derived neurotrophic factor (GDNF) occurs in AS-Exos in reduced amounts [71].

#### 3.1.4. Functions of miRNAs as the AS-Exos Cargo

miRNAs constitute one of the types of AS-Exos cargo released during homeostasis and pathological processes in the CNS. The miRNAs secreted by activated astrocytes may be used as potential biomarkers of inflammation [72]. They can also serve as therapeutic targets for neurological diseases because of their ability to influence cellular functioning by targeting and therefore inhibiting the expression of specific genes [73]. The activity of specific miRNAs secreted in AS-Exos is presented in Table 1.

### 3.2. Characteristics of Exosomes Secreted by ECs

#### 3.2.1. Involvement of Exosomes Secreted by ECs in the BBB Regulation

EC-derived exosomes (EC-Exos) constitute another important way of communication between cells in the NVU. In the study concerning EVs derived from ECs, it was determined that two subsets of EVs are released by these cells. The population of smaller EVs fitted in the size range for exosomes (30–150 nm) and the larger ones were 150–300 nm in diameter and fit in microvesicle dimensions. These two populations despite their uniform origin show distinct properties and different engagement in the BBB regulation in the conditions of induced neuroinflammation. However, the common phenomenon was the increase in concentration in both populations of EVs in the reaction to inflammation. The ability of EVs from ECs to regulate the passing of Th1 and Th17.1 through the BBB was tested on an in vitro BBB model. The results showed that the larger EVs intensified this process in contrast to smaller EVs which did not have such properties. The results from the test on the in vivo EAE model where inflammatory subsets of EVs were utilized revealed that large EVs contributed to the worsening of EAE mice condition, and conversely in the case of small EVs, it was improved. The protective role of small EVs is additionally confirmed by their ability to convert the phenotype of ECs into anti-inflammatory [80]. In another study where ECs were activated by TNF, the protein load present in their exosomes and microvesicles was analyzed. Despite the fact that the same 366 proteins were identified in both of the EV types, there were differences in the cargoes of microvesicles and exosomes. The proteins specific for exosomes constitute tetraspanins, adhesion proteins, histones (HIST1H1D and HIST1H1B), ribosomal proteins, and those engaged in exosome biogenesis (SDC4, SDCBP, VPS4A, and VPS37C). Proteins uniquely present in microvesicles are those related to mitochondria (SOD2, MRPS22, and ATP5A1), cytoskeleton (TUBA1C, TUBB2B, and ACTR2), and those taking part in protein degradation (UBQLN4, PSMC1, and PSMC6). These results indicate that microvesicles’ contents reflect the cytoplasmic contents of the cell of origin which do not take place in the case of exosomes [81]. EC-Exos contributes to maintaining the BBB tightness also by preventing TJ proteins from damage caused by apoptosis. Acting by elevating the Bcl-2/Bax ratio EC-Exos leads to apoptosis inhibition, and by targeting LC3B, the autophagy is halted. The permeability of the BBB is also preserved due to the ability of EC-Exos to inhibit the activity of MMP-9 and maintain ZO-1 and claudin-5 in TJs [82].

#### 3.2.2. EC-Exos Role in Neurodegenerative and Neuroinflammatory Conditions

Neuroprotection along with the prevention of neurodegeneration constitutes another important area of EC-Exos activity. The study on stroke utilizing the ischemic rat models pointed out the application of EC-Exos in axon growth enhancement. EC-Exos isolated both from rats experiencing ischemia and from nonischemic animals proved their ability to reach axons and increase the number of functional miRNAs. Additionally, EC-Exos suppressed the activity of proteins involved in the inhibition of axon’s outgrowth. Worth noticing is the fact that EC-Exos obtained from ischemic rats acted in a more intensified way suggesting not only the protective function but also regenerative [83]. Another study on rats concerning ischemia/reperfusion injury confirms the role of EC-Exos in neuronal recovery. The transcriptome analysis of motor cortex samples revealed that the application of EC-Exo after injury led to an improvement in synaptic transmission and plasticity. Additionally, dendrites were reported to increase their length. Because of these processes, the motor functions could be restored [84]. Huang R et al. emphasize in their study the role of EC-Exos in preventing apoptosis and promoting angiogenesis [85].

EC-Exos may also find application in the context of another neurodegenerative condition which is AD. In one of the AD prevention strategies, the accumulation of Aβ is removed by the P-glycoprotein (P-gp) transporter present in the BBB. During the cause of AD, P-gp is not expressed as efficiently as in homeostasis, and therefore, ways of increasing its amount are desirable. One of them may constitute EC-Exos delivering P-gp to the ECs in the BBB which was presented in the study of Pan J et al. This method led to the increase in P-gp expression and reduction in Aβ amounts, therefore improving the cognitive condition of tested mice [86].

#### 3.2.3. Functions of miRNAs as the EC-Exos Cargo

It was proven in a few studies concerning EC-Exos that their effectiveness is owed to the activity of miRNAs that are part of their cargo. In the case of emerging neurodegenerative conditions such as ischemia or AD, changes in exosomal miRNA were observed which indicates their involvement in the cellular defense response [87]. miRNAs present in EC-Exos display beneficial properties towards both neurons and Ecs, making them useful agents in maintaining CNS homeostasis. The activity of specific miRNAs secreted in EC-Exos is presented in Table 2.

## 4. Conclusions and Future Perspectives

The presence of the BBB in CNS provides protection and maintenance of homeostasis. The main component of the BBB which are ECs ensure its tightness due to the structural adjustments. TJs prevent the entry of larger molecules into the bloodstream in the CNS, whereas transporters pass them in a controlled way. Astrocytes constitute another important element of CNS which serves as a modulator of its environment. These cells are able to receive signals from both neurons and ECs which allows them to have a mutual influence. The communication between astrocytes and ECs is crucial in the context of the BBB regulation. The paracrine secretion of signaling molecules is the most thoroughly studied way of information exchange between these two types of cells. Multiple new studies show that every type of cell has the possibility to secrete exosomes which may serve as an alternative way of communication. The current knowledge about AS-Exos emphasizes their role in neuroprotection, reduction in inflammation, neuronal regeneration, and remyelination. EC-Exos are involved in modulation and maintaining the tightness of the BBB, and similarly to AS-Exos, have anti-inflammatory, neuroregenerative, and neuroprotective properties. Taking into consideration the importance of mutual cooperation between astrocytes with ECs in CNS and the vital role of their exosomes, it is worth exploring the involvement of these vesicles in communication between these cells. The knowledge about the influence of AS-Exos on ECs and EC-Exos on astrocytes can form the basis for the future application of these exosomes in the treatment of neurodegenerative and neuroinflammatory diseases. Nowadays, many clinical trials focus on the utilization of exosomes in the treatment of CNS diseases. Particularly, engineered exosomes have the potential to be diagnostic and therapeutic tools since they can selectively reach their targets [88]. As examples may serve allogenic adipose MSC-derived EVs in AD (Clinical Trial Number: NCT04388982), iPSC-derived EVs in refractory focal epilepsy (NCT05886205), and allogenic MSC-derived EVs enriched with miR-124 in acute ischemic stroke (NCT03384433) [89].

## Figures and Tables

**Figure 1 ijms-26-04676-f001:**
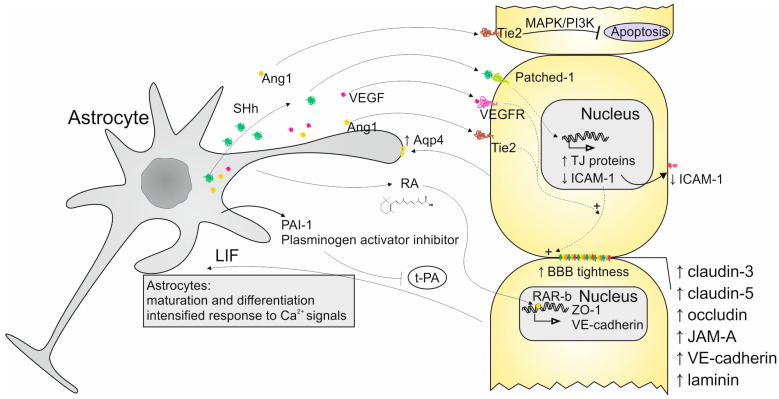
Secretory crosstalk between astrocytes and ECs and its role in the BBB support. Arrows indicate upregulation or downregulation of proteins secretion.

**Figure 2 ijms-26-04676-f002:**
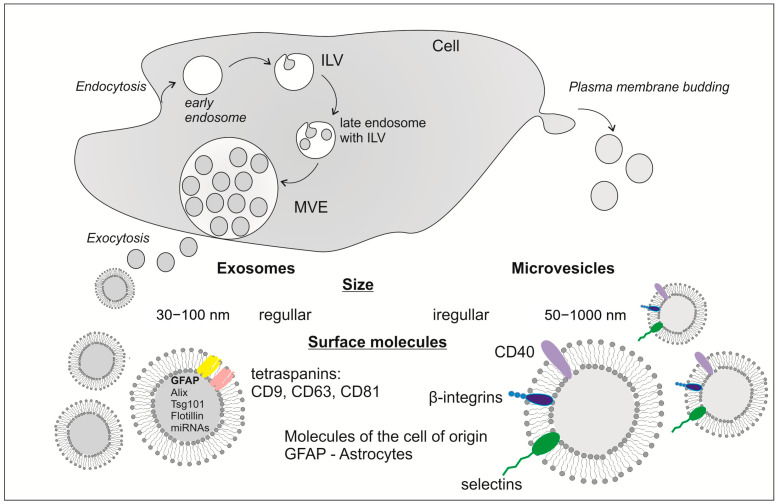
Comparison of exosomes and microvesicle formation, release, and characteristics.

**Figure 3 ijms-26-04676-f003:**
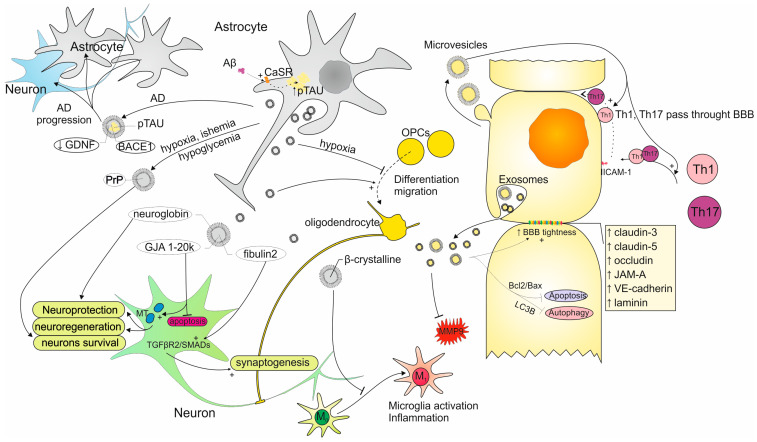
The importance of exosomes in the interactions between cells in the brain tissue. Arrows indicate upregulation of proteins secretion.

**Table 1 ijms-26-04676-t001:** The activity and targets of selected miRNAs from AS-Exos in CNS environment.

miRNA	Potential Targets	Possible Effects	References
miR-520d-3p	*BACE*	β-amyloid protein regulation	[72]
miR-29a	*APP*		
miR-let-7d miR-30d miR-31-3p miR-93-3p miR-145-5p	GFAP, aquaporin, vimentin, and amyloid precursor protein	Involvement in neurological disorders and brain trauma	[72]
miR-26a	*CTDSP2*	Neurogenesis	
	*GSK-3β*	Regeneration of axon	[73]
	*PTEN*	Intensification of neurite outgrowth	
miR-378a-5p	*NLRP3*	Reduction in pyroptosis and neuroinflammation	[74]
miR-34c	*TLR7*	Reduced neuronal damage after ischemia/reperfusion	[75]
miRNA-17-5p	*BNIP-2*	Protection of the brain from hypoxic–ischemic damage	[76]
miR-182-5p	*Rac1*	Inhibition of neuroinflammation after ischemic stroke	[77]
miR-190b	*Atg7*	Suppression of neuronal apoptosis and inhibition of autophagy caused by glucose deprivation	[78]
miR-873a-5p	ERK, NF-κB, and p65	Enhancing microglia conversion into the M2 phenotype	[79]

**Table 2 ijms-26-04676-t002:** The activity and targets of selected miRNAs from EC-Exos in CNS environment.

miRNA	Potential Targets	Possible Effects	References
miR-27a miR-19a miR-195 miR-298	*NCAM1, SEMA6A,* and *SEMA7A* *PTEN* *RTN4* *RHOG* and *RHOA*	Inactivation of proteins suppressing axonal growth	[83]
miR-126-3p	*PIK3R2*	Protection from the BBB damage	[84]
	*SDF-1*	Promotion of angiogenesis Enhancement of neurite outgrowth	
miR-122-5p, miR-409-3p	*Sbk1 and Syne2*	Cell proliferation and migration	[87]
miR-412-5p	*Larp1*	Vasculature formation	
miR-379-5p	*Krt26*	Regeneration of nerve fiber	
miR-494-3p	*Hsf2*	Apoptosis	
miR-127-3p	*Acat1*	Cellular adhesion	

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
