# Peer review of "The Potential Role of Exosomes in Communication Between Astrocytes and Endothelial Cells"

_ijms, 2025, doi:10.3390/ijms26104676_

Round 1

Reviewer 1 Report

Comments and Suggestions for Authors

Attached

Comments on the Quality of English Language

There are some grammatical errors and the sentences needs to be reframed for better presentation.

Author Response

Comments 1:

The fullforms of the abbreviations should be included at first instance. For example, Zo-1, VE-cadherin,
EAE 

Response 1: 

Dear Reviewer, thank you for pointing this out. The full forms of the abbreviations have been applied and highlighted in lines 57, 109, 122, 123, 130, 172, and 217.

Comments 2:

The introduction could be curtailed into quick background about the blood brain barrier and focusing
more on the importance of crosstalk between astrocytes and endothelial cells which will be of interest to
the readers. It needs to include a draft explaining the paracrine and autocrine signalling pathways and
their role in the crosstalk between astrocytes and endothelial cells.

Response 2: 

Thank you for this comment, however, the introduction presents the background about the blood-brain barrier in a concise and precise manner simultaneously keeping the paragraph 1.1. short and containing information about its role, functions, importance, and structure. The paragraphs 1.2. and 1.3. serve as the opening to the topic of the crosstalk between astrocytes and endothelial cells which is presented in paragraph ‘2. Interactions between ECs and astrocytes’. Since paragraph 2. is placed right after the brief introduction, we don’t think that it is necessary to mention this topic in the Introduction. The paracrine signaling pathways are also mentioned in paragraph 2. (e.g. Hedgehog pathway, MAP kinase pathway, PI3-kinase pathway et cetera). We added the new figure (Figure 3.), which includes information about autocrine signaling using exosomes.

Comments 3:

Line 110-The condition of BBB appears to improve after the APOE4 knockout is not required since it
does not give any additional information. As stated earlier only the molecular mechanism vital to
understand the crosstalk between astrocytes and endothelial cells should be elaborated in the
introduction.

Response 3: 

We deleted the sentence ‘The condition of BBB appears to improve after the APOE4 knockout’ from the line 110.

Comments 4:

The manuscript needs to be improvised for the language. For instance, ‘Thanks to the presence of the
Patched-1 receptor on the surface of ECs, astrocytes can initiate the signal transduction by production of
its agonist Sonic Hh (SHh)’. Can be rewritten as ‘astrocytes can initiate the signal transduction by
production of its agonist Sonic Hh (SHh) due to the presence of Patched-1 receptor on EC surface’. I
encourage the authors to use English language tools like Grammarly. 

Response 4: 

Thank you for your suggestion. The sentence has been rewritten in lines 118-120. Also, the language improvement has been applied to the whole manuscript.

Comments 5:

It is essential to include a graphical figure explaining the crosstalk between astrocytes and endothelial
cells.

Response 5:

According to your suggestion, we inserted the Figure 1.

Comments 6:

The section 2 elaborates on different mechanisms involved between astrocytes and endothelial cells.
However, the role of inherent exosomes with specific marker proteins secreted by either the astrocytes
and endothelial cells involved in these mechanisms is missing. Moreover, the connectivity between
different mechanisms in missing. For instance, there was a jump from hedgehog pathway to ICAM-1 and
then to VEGF suddenly onto proteins involved in building TJs and transporters. The section should be
reorganized to include connectivity between different mechanisms inclusive of the role of exosomes.

Response 6:

As you noticed, the role of inherent exosomes with specific marker proteins secreted by either the astrocytes and endothelial cells involved in these mechanisms is missing, because the section 2 concerns solely the interactions between endothelial cells and astrocytes, not exosomes secreted by these cells. The role of inherent exosomes from astrocytes and endothelial cells is described in the section 3 ‘Exosomes as the strategy of communication between astrocytes and ECs’. We agree with the comment about the need to improve connectivity between different mechanisms, therefore we reorganized the section 2.

Comments 7:

In section 3, was there a specific reason to include only CD81, CD9, and CD63 markers expressed on
exosomes? The exosomes from the tight cerebral junction have additional markers including but not
limited to L1CAM, CD13, Myelin, HSP60, HSP70, ESCRT-related components. Please elaborate on the
significance of these markers too.

Response 7:

In section 3, the 3 markers of exosomes are mentioned because they are the major components of exosomes. Now we added this information to the text. This paragraph consists of a few characteristics of exosomes in general and it is not about exosomes from the tight cerebral junction, therefore we decided not to elaborate on the significance of  L1CAM, CD13, Myelin, HSP60, HSP70, ESCRT-related components.

Comments 8:

Section 3 is more general introduction of exosomes, the section should include the advantage of using
exosomes over microvesicles and compare them for their physicochemical properties, including cargo
load, charge, surface markers, and particle size.

Response 8:

Lines 189-193 ‘ The number of research on exosomes is increasing year by year and overtake the number of publications about microvesicles which can be explained by the exosomal properties regarding use in potential therapies, the route of biogenesis, successful delivery to cells, and increased oral bioavailability [56].’ contain the reasons behind the increased focus on studies concerning more often exosomes than microvesicles and simultaneously shows the advantage of using
exosomes over microvesicles. The comparison of their physiochemical properties is present on the Figure 2.

Comments 9:

Section 3, describes different exosomes and their content which may lead to efficacy. However, it is vital
to incorporate the mechanism of each effector molecules loaded within exosomes leading to the effect.

Response 9: 

Section 3, especially subsections 3.1. and 3.2., contain information about the mechanisms of each effector molecules loaded within exosomes leading to the effect. Additionally, Table 1. and Table 2. contain specific information about the activity and targets of selected miRNAs which are the content of exosomes. The examples in the text are: (lines 226-229) ‘AS-Exos are also engaged in the formation of synapses by activating transforming growth factor β (TGF-β) signaling with the use of fibulin-2. This protein is present in AS-Exos in significantly higher amounts in comparison to other exosomes [61].’, (lines 239-242) ‘In the study with a mouse model of experimental autoimmune encephalomyelitis (EAE) it was pointed out that α-B-crystalline protein present in AS-Exos suppressed microglial activation and subsequent inflammation [63].’, (lines 247-251) ‘The research performed by Chen W et al. indicates that astrocytes participate in neuronal repair by the release of exosomes with enclosed gap junction alpha 1-20 k protein (GJA1-20 k). This protein is involved in apoptosis suppression and enhancing mitochondrial activity which contributes to neuronal protection and regeneration [64].’, et cetera.

Comments 10:

Incorporate a table including application of exosomes in different neuronal diseases which has hampered
crosstalk between astrocyte and endothelial cells.

Response 10: 

Thank you for this comment. However, the information about the application of exosomes from astrocytes and endothelial cells in various neuronal diseases is already present in the text. We believe that doubling the information is not necessary since paragraphs 3.1.3 and 3.2.2 cover this topic comprehensively.

Comments 11:

The concluding remarks and future perspective should include authors perspective on commercial and
clinical translation of the exosomes for neuronal disorders quoting examples which are under clinical
trials, if applicable.

Response 11: 

As you suggested, we added the examples of commercial and clinical translation of the exosomes for neuronal disorders in ‘Conclusions and future perspectives’.

Reviewer 2 Report

Comments and Suggestions for Authors

This review article discusses the role of extracellular vesicles (EVs), particularly exosomes, in communication between astrocytes and endothelial cells within the context of the blood-brain barrier (BBB). The manuscript first outlines the physiological contributions of astrocytes and endothelial cells to BBB maintenance, followed by a discussion on intercellular communication mechanisms, including tight junctions and signaling. The latter half of the review focuses on EV-mediated communication between these two cell types, emphasizing their relevance in neuroinflammation and neurodegeneration.

The review is timely and well-written, and with a few minor corrections, it can be strengthened further:

  1. A major concern is that while the review presents insights into the role of EVs from astrocytes and endothelial cells in neuroinflammation and neurodegeneration, it offers limited or no discussion on whether these EVs directly affect BBB permeability or integrity. The authors should include whether there are any reports of EVs from astrocytes or endothelial cells directly affecting BBB permeability or maintenance. From the current review, there appears to be very limited, if any, discussion on this. If such data is lacking in the current literature, but the title and abstract should be revised accordingly, so as not to mislead readers into expecting a focused review on EV-mediated BBB regulation.
  2. Lines 161–170, which discuss exosome biogenesis, the ESCRT machinery, and MVB fate, lack appropriate referencing. The authors should include citations to support these.
  3. Line 197- “The synthesis of knowledge regarding the exosomal way of acting….” This line is not clear, what it’s trying to convey.
  4. Line 208 – the report that is described, does it mention that the uptake of AS-Exos is specific to neurons? If so, compared to what cell types? The authors can mention that in more detail.
  5. Line 291 – What is EAE? Not mentioned.
  6. While the EV biogenesis figure is appreciated, similar figures are widely available in the literature. Including an additional figure illustrating how astrocytes and endothelial cells support the BBB and how their EVs impact neuronal functions would add clarity and give the review a unique and stronger identity.

Author Response

Comments 1:

A major concern is that while the review presents insights into the role of EVs from astrocytes and endothelial cells in neuroinflammation and neurodegeneration, it offers limited or no discussion on whether these EVs directly affect BBB permeability or integrity. The authors should include whether there are any reports of EVs from astrocytes or endothelial cells directly affecting BBB permeability or maintenance. From the current review, there appears to be very limited, if any, discussion on this. If such data is lacking in the current literature, but the title and abstract should be revised accordingly, so as not to mislead readers into expecting a focused review on EV-mediated BBB regulation.

Response 1: 

The review contains discussion on the influence of the EVs from endothelial cells on BBB permeability, and integrity in the paragraph 3.2.1. ‘Involvement of exosomes secreted by ECs in BBB regulation’. In case of exosomes from astrocytes, there is no studies on this topic and this area of knowledge is still not explored. However, since the endothelial cells constitute the integral part of BBB, the influence of astrocytes on BBB is mentioned in the section 2 ‘Interactions between ECs and astrocytes’. As it was written in lines 198-204, the knowledge about the communication between astrocytes and endothelial cells with the use of exosomes is restricted. Following your suggestion, we changed the title to ‘The potential role of exosomes in communication between astrocytes and endothelial cells’.

Comments 2:

Lines 161–170, which discuss exosome biogenesis, the ESCRT machinery, and MVB fate, lack appropriate referencing. The authors should include citations to support these.

Response 2: 

The additional citation was inserted in lines 170 and 175.

Comments 3:

Line 197- “The synthesis of knowledge regarding the exosomal way of acting….” This line is not clear, what it’s trying to convey.

Response 3: 

The purpose of these sentence was to stress the importance of the gathered knowledge regarding exosomes secreted from astrocytes and endothelial cells in the further research on them. Following your suggestion, the sentence has been clarified.

Comments 4:

Line 208 – the report that is described, does it mention that the uptake of AS-Exos is specific to neurons? If so, compared to what cell types? The authors can mention that in more detail.

Response 4: 

Thank you for these comment. Following your suggestion, we elaborated on the specific to neurons uptake of AS-Exos. However, this study focuses on the co-culture of only astrocytes with neurons. There were no other cells types described, therefore the results show that AS-Exos are internalized by neurons with which astrocytes had contact. This study will not provide a comparison of uptakes by other cells and to our best knowledge similar studies done on other cells types are not available.

Comments 5:

Line 291 – What is EAE? Not mentioned.

Response 5: 

The EAE abbreviation is firstly mentioned and elaborated in the 239 line therefore we assumed that the repeated explanation is not needed.

Comments 6:

While the EV biogenesis figure is appreciated, similar figures are widely available in the literature. Including an additional figure illustrating how astrocytes and endothelial cells support the BBB and how their EVs impact neuronal functions would add clarity and give the review a unique and stronger identity.

Response 6: 

According to your suggestion, we inserted the new figures (Figure 1. and Figure 3.)